

# ssNMRlib: a comprehensive library and tool box for acquisition of solid-state NMR experiments on Bruker spectrometers

Alicia Vallet[1], Adrien Favier[1], Bernhard Brutscher[1], and Paul Schanda[1]

[1]Univ. Grenoble Alpes, CEA, CNRS, Institut de Biologie Structurale (IBS), 71, Avenue des Martyrs, F-38044 Grenoble, France

**Correspondence:** Alicia Vallet (alicia.vallet@ibs.fr), Paul Schanda (paul.schanda@ist.ac.at)

**Abstract.** We introduce ssNMRlib, a comprehensive suite of pulse sequences and jython scripts for user-friendly solid-state NMR data acquisition, parameter optimization and storage on Bruker spectrometers. ssNMRlib allows the straightforward setup of even highly complex multi-dimensional solid-state NMR experiments with a few clicks, from an intuitive graphical interface directly from the Bruker Topspin acquisition software. ssNMRlib allows the setup of experiments in a magnetic-field independent manner, and thus facilitates the workflow it in a multi-spectrometer setting with a centralized library. Safety checks furthermore assist the user in experiment setup. Currently hosting more than 140 1D to 4D experiments, primarily for biomolecular solid-state NMR, the library can be easily customized and new experiments are readily added as new templates. ssNMRlib is part of the previously introduced NMRlib library, which comprises many solution-NMR pulse sequences and macros.

## 1 Introduction

Nuclear magnetic resonance is arguably the most versatile spectroscopic technique, with applications ranging from studies of molecules in the solid, liquid or gas phases to complex materials and even entire organisms. The versatility of NMR spectroscopy is rooted in the countless opportunities to manipulate nuclear spins. Nowadays, the NMR spectroscopist can choose from a large number of highly specialized pulse sequences to obtain answers to specific questions related to molecular structure or dynamics. Most NMR experiments contain a number of building blocks used for transfer of coherence, for observation of chemical shifts in directly observed or incremented indirect dimensions, and blocks monitoring spin evolution due to relaxation, or spin evolution under the action of (dipolar or CSA) recoupling sequences. The quality of NMR data critically depends on the precise setting of many parameters related to each of these building blocks. As a consequence, setting up an NMR experiment that yields the best possible data is often a complex time consuming procedure even for specialists. Accordingly, valuable experiment time is often spent on optimizing parameters. Furthermore, keeping track of already optimized parameters in one experiment for use in another experiment that uses the same building blocks, is an error-prone process. In facilities with multiple spectrometers, it is often a concern how the pulse sequences are centralized, and how one can obtain a set of parameters that safely works for a given pulse sequence on a given spectrometer, or how to transfer an acquisition parameter set from one spectrometer to another.





Here, we introduce a library of pulse sequences, scripts and intuitive graphical-interface based setup routines for solid-state NMR experiments on Bruker spectrometers, ssNMRlib. This library is built upon NMRlib, recently developed for biomolecular solution-state NMR (Favier and Brutscher (2019)), and has numerous solid-state NMR specific features that greatly facilitate the key tasks of the experimentalist, from parameter optimization and storage of optimized parameters, to rapid and easy setup of complex pulse sequences, centralization of pulse sequences and user-friendly storage of acquisition parameters for

later use in publications or laboratory notebooks. ssNMRlib is open to many kinds of applications. We have included so far more than 140 pulse sequences for biomolecular MAS experiments including heteronuclear multi-dimensional (1D-4D) correlation spectroscopy for resonance assignment, structure determination and dynamics studies, using $^{1}$H, $^{13}$C and $^{15}$N detection; $^{31}$P and $^{19}$F detection are currently being added. The ssNMRlib concept is fully open to accommodate many kinds of pulse sequences, and thus not limited to biological ssNMR. The possibility that the user includes new experiments into the

library – which then can also benefit from the automatic setup routines – is a central idea to ssNMRlib. The integration into NMRlib offers many additional scripts and macros e.g. for data processing or python-based fitting and printing. In this article we describe the key philosophy and concepts of the library, the workflow, and exemplify its use with several applications.

### 1.1 Overview of the aims of ssNMRlib

The central design ideas behind ssNMRlib are summarized in Figure 1A. We sought to create a tool that accesses a central

library of scripts and pulse sequences, which allows setting up experiments independently of the magnetic field strength, and which takes into account the installed probe head. Parameters specific to a given machine and probe head (such as Larmor frequencies and RF power levels) are loaded by a centralized script that reads spectrometer- and probe-specific values.

Furthermore, as many pulse sequences use common building blocks, e.g. for coherence-transfer steps, an important aim of ssNMRlib is to allow the user to optimize the relevant parameters once, ideally in an automated manner. The optimized

parameters are saved without the need for additional user interference (but with the possibility of editing all parameters). These optimized parameters are then automatically read into more complex experiments, where these building blocks are needed. To facilitate the setup further, ssNMRlib proposes reasonable starting values, e.g. for cross-polarization transfers, taking into account the (automatically retrieved) MAS frequency. The transfer of parameters between different experiments (e.g. the retrieval of CP parameters to be used in complex 2D/3D/4D experiments) is facilitated by a common nomenclature of

parameters, as outlined in Table S1 in the Supplementary Information.

Pulse sequences in ssNMRlib are written such that the power levels relevant for pulse sequence elements are entered as constants (*cnst*) in units of Hertz or kilohertz, rather than in Watt or dB. Field strength in kilohertz is the relevant quantity for essentially all transfer steps, e.g. for Hartmann-Hahn cross-polarization conditions (Hartmann and Hahn, 1962), for many common recoupling schemes such as HORROR (Nielsen et al., 1994), DREAM (Verel et al., 1998), PAIN (Lewandowski et al.,

2007), DARR (Takegoshi et al., 2001) or MIRROR (Scholz et al., 2008), as well as for decoupling. Handling power levels in kHz units rather than Watt/dB is, therefore, more convenient and intuitive. The conversion to Watt units is done internally within the pulse sequences, using the hard-pulse calibration. We find that on temporary hardware with linearized amplifiers, one can calculate accurate power levels in Watt from a single 90-degree pulse calibration.





**MAGNETIC RESONANCE**
Discussions



**Figure 1.** Overview of the functionlities of ssNMRlib. (A) Main ideas considered in the design of ssNMRlib. (B) Part of the current menu structure of ssNMRlib, here focussing mainly on [1]H-detected experiments. Highlighted are routines for calibration (orange/red) and 4D backbone assignments experiments (blue, right). Helper scripts are shown in purple (bottom).

In all cases where selective pulses are used, e.g. in CO-CA INEPT transfers or homonuclear [13]C decoupling in indirect dimensions, the pulse power levels are calculated directly within the pulse sequences using the desired excitation band width (in ppm), i.e. at each $B_0$ field strength the correct power level is calculated. The user does not need to take care of the power-level calibration of any shaped-pulse.

   An important further consideration when designing NMRlib is the possibility to add new experiments to the library, which
then are accessible as a button that performs the setup of the experiment. To include a novel experiment to ssNMRlib, buttons are available which create the template and setup scripts from an experiment that has been collected. The newly created setup



routine is available as a button in the ssNMRlib browser window (see below). The organization of buttons within the NMRlib GUI tree structure is fully customizable.

We have, moreover, implemented a safety check routine, which verifies whether the chosen RF parameters (pulses, ramps,
parameter optimizations, vdlist, delays etc.) are within the probe-specific limits.

## 2  Implementation of ssNMRlib

NMRlib is an add-on in the Bruker Topspin software, and has been tested currently on Topspin 3.2 to 3.6 and Topspin 4, on Linux host computers. ssNMRlib is part of the NMRlib library, and is installed at the same time as NMRlib. The NMRlib GUI window is launched directly from a button within the Topspin software (see Figure 2A). NMRlib comprises a set of jython
scripts, pulse sequences, selective-pulse and ramp shapes, and acquisition lists (*vdlist, vclist* etc), which can be saved either locally on one spectrometer, or on a centrally mounted disk (which may be on one spectrometer), to which all spectrometers access. The latter is very useful to ensure that a whole facility uses the same experiments. The individual experiments (pulse programs, one jython script for acquisition, one for processing parameters and one for the security check) are organized in a directory tree structure and buttons in the GUI windows allow navigating through these directories. The organization of the
experiments can be fully customized, and new experiments can be added to any location within the file structure. The library also contains all the shape files for selective pulses, amplitude ramps etc, and decoupling sequences used by ssNMRlib, as well as a file required for safety checks (see below). On each spectrometer, the files with the correct routing need to be available, e.g. for $^1$H, $^{13}$C or $^{15}$N detection experiments. The principal elements of the *prosol* table (power levels of the different channels) need to be up to date. These latter instrument-specific files, together with the general, non-instrument-specific scripts and the
pulse sequence contain everything needed to run ssNMRlib on a given instrument.

Part of the current ssNMRlib experiment file tree is shown in Figure 1B. Example screenshots of the NMRlib windows are shown in Figure 2A. Currently, the library is organized to have in the top level branches (buttons) for setup experiments (KBr, adamantane), a main branch containing $^1$H-, $^{13}$C- or $^{15}$N-detected experiments, as well as buttons leading to scripts, including those that allow adding/removing templates to/from the library, saving acquisition parameters or retrieving them from
a previous session. In addition, the NMRlib window also contains an "Editor" button to navigate directly to the folder structure that contains the underlying scripts and pulse sequences pertaining to a given NMRlib window ("pp" (pulse sequences),"py" (jython scripts), "vc" and "vd" (lists)). Furthermore, a "Save" button allows saving an experiment in order to create a template from an experiment that has been acquired (see section 2.4).

The tasks performed by a typical script in ssNMRlib which sets up an experiment are shown in Figure 2B. It starts by reading a file that contains the probe/amplifier routing. The further steps of the setup script load the pulse sequence, adjust the dimensionality, read the current MAS frequency, set pulse lengths (either from prosol or if available from a previous calibration), calculate acquisition parameters based on the MAS frequency and e.g. Hartmann-Hahn condition or automatically retrieve previously optimized parameters, calculate the decoupling power levels from kHz values and load processing parameters. Lastly,





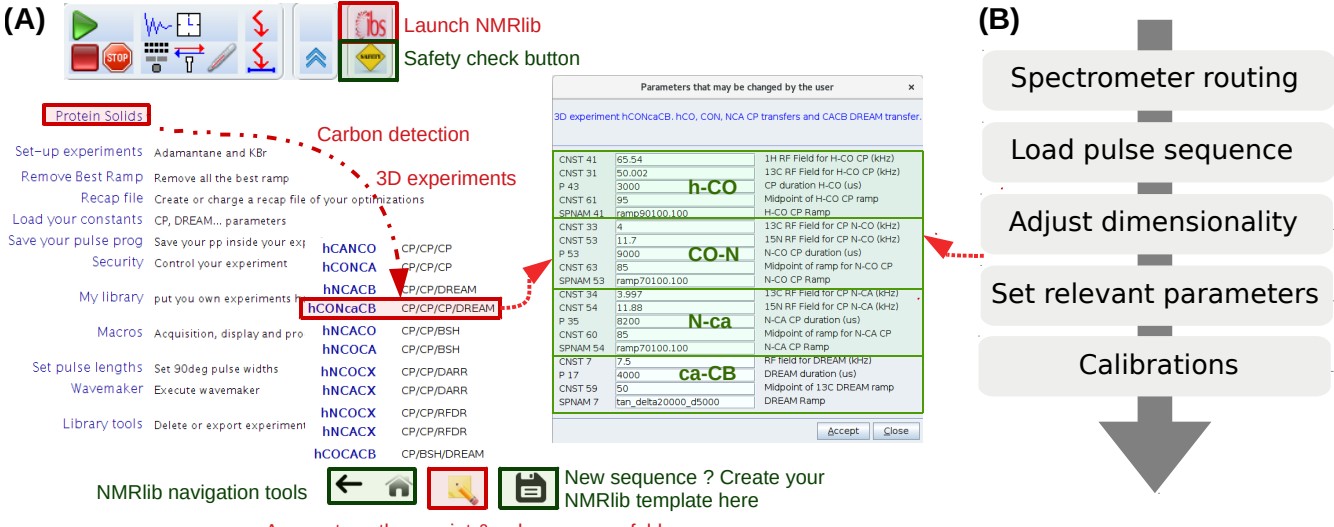

**Figure 2.** (A) ssNMRlib windows and example workflow. An icon with the logo "ibs" in Topspin (top) launches the main GUI window (lower left). Here the navigation to a carbon-detected 3D hCONcaCB experiment is exemplified. After clicking the corresponding experiment button, a window with the relevant parameters opens, including reasonable propositions or (if available) pre-calibrated values of the most relevant acquisition parameters (right). The buttons on the bottom of the window allow navigation within the ssNMRlib file structures, access the underlying scripts in a file browser window, or saving the current experiment as a template. (B) Flow chart of the steps performed by the underlying ssNMRlib script when clicking an experiment button. The spectrometer routing files (one per detected nuclei) are a parameter set, obtained by saving (*wpar*) the parameter set of a high-dimensional experiment (typically 4D) with the correct routing.

the script opens a GUI window that summarizes the most important parameters (Figure 2, center). The user may manually modify parameters at this step. Of course, all parameters can also be changed at any point in the *AcquPars* tab in Topspin. Another added value of the ssNMRlib software is that all the parameters of an experiment are automatically checked to insure the probe safety before launching an experiment.

## 2.1 Work flow with ssNMRlib

The typical workflow with ssNMRlib can be described as follows. After setup of the sample spinning and probe tuning, possibly with adjustment of the magic angle and shims (templates for KBr acquisition (Frye and Maciel, 1982) and adamantane $^{13}$C observation are included in ssNMRlib), the user typically wants to calibrate 90 degree pulses. A number of automated routines, described below, allow pulse calibration. The optimized values are stored and then used for the setup of all other experiments. In a similar manner, further calibration experiments e.g. for cross-polarization or scalar-coupling based transfer are available

with buttons in the calibration menu, and their optimized values (power levels, shape parameters, durations etc.) are likewise stored by ssNMRlib. All optimized parameters of hard pulses and transfer parameters are stored in a dedicated text file. This





file can also be re-imported into ssNMRlib at a later point; this feature is handy e.g. if one interrupts a measurements session and continues at a later time point, starting with the previously optimized acquisition parameters.

Once all the required transfers and pulses have been optimized, clicking a button for a specific experiment (e.g. a 2D, 3D or 4D correlation experiment) executes a script that automatically reads the optimized parameters used in all the transfer steps, calculates power levels and shapes, and sets up the particular experiment. These steps are described in the following sections.

### 2.1.1 Pulse calibration

A number of experiments for 90° pulse calibrations are available, e.g. a nutation experiment with a direct hard pulse excitation or with CP or INEPT transfer schemes. Figure 3 exemplifies a $^1$H pulse calibration with a $^1$H detected hNH experiment, typically used for proteins at fast MAS. The underlying python script loads the instrument-specific parameter set (routing, power levels, pulse lengths etc) and pulse sequence, and sets up a parameter optimization (*popt* in Bruker language) protocol using pre-defined spectral ranges to be observed (e.g. amide $^1$H frequency range) and the range of values used for the parameter optimization, in this case $^1$H pulse length. This range is based on typical values for the present probe, which are retrieved from the *prosol* table. At the end of the parameter optimization, the pulse duration is retrieved automatically (e.g. from finding the zero-crossing of the integrated spectral intensities). The value of the 90 ° pulse is stored, and can be inspected and modified within the NMRlib window any time (Figure 3).

### 2.1.2 Automatic calculation of power levels and shapes in ssNMRlib

With calibrated 90 ° pulse durations and power levels, and a linearized amplifier, all power levels of decoupling and recoupling pulses and shapes, as well as selective pulses can be calculated automatically. ssNMRlib does not require the user to do any calculation in Watt or dB units, but wherever possible makes calculations from user-specified values in kilohertz units. Furthermore, selective pulses, e.g. REBURP, EBURP (Geen and Freeman, 1990) or ISNOB (Kupce et al., 1995) are directly calibrated within the pulse sequence, based on the known characteristics of a given pulse shape, as well as the desired excitation band width.

Figure 4 shows pulse sequence snippets that illustrate how these calculations are done in practice. The calibrated 90° hard pulse durations and power levels can be translated to the RF field strengths (assigned to variables *cnst1*, *cnst2*, *cnst3*, *cnst4* for $^{13}$C, $^1$H, $^{15}$N and $^2$H nuclei, respectively) that correspond to the power levels at which these pulses have been calibrated. Selective pulses are then calculated for a given shape, excitation band width, spectrometer frequency and the hard-pulse calibration. The use of the spectrometer frequency ensures that the parameters are calculated correctly for any field strength. Possible frequency shifts from the carrier to the excitation band are calculated in a spectrometer-independent manner (in ppm).

The calculation of power levels for cross-polarization or other amplitude ramps is illustrated in Figure 4C, D. The power level is specified by the user in kilohertz as constants (*cnst*) in Topspin, and internally translated to Watt, using the 90° hard





**Figure 3.** Procedure for hard-pulse calibration in ssNMRlib, exemplified for the case of the $^1$H pulse, detected via a $^{15}$N-filtered CP-based hNH experiment. A popup window asks whether the calibration shall be used as the basis for all later calculations (of selective pulses, CP transfers and decoupling, expressed later in kHz; button named "Before"), or whether it is a re-calibration/verification of the pulse, that shall not impact the calculation of CP power levels (button named "After"). Different CP ramps are available, and reasonable values are proposed, based on the MAS frequency and Hartmann-Hahn match conditions. Lastly, a popup window summarizes the most relevant acquisition parameters before the *popt* procedure is launched. The optimized value is automatically stored; the cogwheel symbol in the ssNMRlib window next to the button allows displaying (and editing) the currently stored optimized value.





```
(A) /********** Calculation of power levels of CPs from  pulse calibrations
    ; get rf fields (in kHz) based on the 90deg pulse calibrations
    ;hard pulse power levels for calibration
    "cnst1=1000/(4.0*p21)" ;13C
    "cnst2=1000/(4.0*p22)" ;1H
    "cnst3=1000/(4.0*p23)" ;15N

(B) /******* selective 13C pulses  *******/
    ; CA pi pulse (off-resonance)
    "p18=1.7/(70.0*bf2/1000000)" /*  ISNOB-2   */
    "spw18=plw2*(pow((p21*2.0/p18)/0.2144,2))"   /* ISNOB-2  */
    "spoff18=bf2*(-(cnst16-cnst17)/1000000)"   /* shift from CO (carrier) to CA */

    ; CO pi pulse (on-resonance)
    "p19=4.875/(70.0*bf2/1000000)" /* REBURP pulse length  */
    "spw19=plw2*(pow((p21*1.97/p19)/0.0798,2))"   /* REBURP power level */
    "spoff19=0"

(C) ; H-C CP, ramp is on H
    "plw31=plw2*(pow((cnst31/cnst1),2))"
    "spw41=plw1*(pow((cnst41/cnst2)/(cnst61/100.0),2))"

(D) ; CO-CA BSH-CP
    "spw38=plw2*(pow((cnst38/cnst2)/(cnst58/100.0),2))"
```

**Figure 4.** Calculation of RF pulse parameters in ssNMRlib sequences for spectrometer-frequency independent setup. (A) The hard pulses (*p21, p22, p23*) are translated to RF field strengths in kilohertz (*cnst1, cnst2, cnst3*). (B) Calculation of selective-pulse parameters exemplified for off-resonance ISNOB2 and an on-resonance REBURP shape, using 70 ppm excitation bandwidth, at a spectrometer operating at a frequency of *bf2* (in MHz). The values of *cnst16* and *cnst17* represent the centers of the CO and CA frequency bands, respectively in ppm units. (C) Calculation of power levels of a cross-polarization, here for a $^1$H-$^{13}$C transfer with an amplitude ramp on the $^1$H channel. The desired RF-field values in kHz in the middle of the ramp are specified in *cnst31* (for $^{13}$C) and *cnst41* ($^1$H); the additional *cnst61* corresponds to the percentage value of ramp in its center, e.g. *cnst61*=85 for ramp that goes from 70 to 100 %. As the power level of the ramp corresponds to the highest value of the ramp, this constant is needed for the calculation, in addition to the midpoint RF field strength. (D) Calculation of amplitude for a ramped transfer on one channel, such as DREAM or BSH-CP. The RF field strength at the mid-point of the ramp corresponds to *cnst38*.

pulse durations and power levels. The use of kHz values allows the user to immediately have a reasonable set of starting values,

based on Hartmann-Hahn matching conditions for CPs, HORROR conditions etc.

Currently, the shape files are not created on the fly (which might be done with Bruker's *WaveMaker* library), but they are stored locally. The user can change the CP ramp file to any desired shape in the *AcquPars* tab. Just like these recoupling sequence elements, the decoupling power levels are calculated from desired RF field strengths (in kHz).

### 2.1.3   Optimization of transfer elements

The calibration of transfer steps, based on e.g. cross-polarization, other dipolar-based transfer such as RFDR, DREAM, BSH-CP, and INEPT-based transfers, follows a similar philosophy, exemplified in Figure 5 for the optimization of a CO-CA BSH-CP (left) and a $^1$H-$^{13}$C CP transfer (right). Dedicated buttons for each of these transfers allow loading calibration experiments, which use the previously optimized hard pulses. A ssNMRlib window summarizes the acquisition parameters, allows choosing





the ramp or the carrier position, and sets up a *popt* optimization protocol. The optimized parameters are stored for later use in
other experiments in ssNMRlib, and can be modified from the NMRlib GUI, and saved to a file. In addition, a function, termed
"*Load constants*" allows retrieving at any point a particular set of parameters, e. g. the cross-polarization power levels, shape
and duration.

ssNMRlib is open to the use of different modes of detection ($^1$H, $^{13}$C, $^{15}$N) within the same session and on the same probe.
For example, one may want to collect a $^{13}$C-detected experiment and a $^1$H-detected one on the same sample and same probe.
Accordingly, calibrations done in one detection mode (e.g. a N-C CP transfer optimized with a $^{13}$C-detected experiment) is
automatically retrieved for a $^1$H-detected experiment that uses this transfer (e.g. a H-N-C correlation experiment).

Of note, ssNMRlib uses a uniform naming convention for acquisition parameters. For example, a common parameter name,
such as *cnst41*, is used for the CP power level of $^1$H in all occurrences of $^1$H-$^{13}$C CPs, and it is different from e. g. the one
used for the $^1$H power in $^1$H-$^{15}$N CPs. This clear naming convention (see Table S1) additionally helps the user to retrieve
parameters – although ssNMRlib is made such that the user does not need to remember those names, due to the automatic
parameter retrieval. It is recommended to use the same naming convention when adding new experiments. A consistent naming
convention is also very useful for safety checks (see section 2.3).

## 2.2 Current content and organization of ssNMRlib

ssNMRlib can be fully customized, i. e. experiments can readily be added, deleted, and the organization can be modified, as
described in section 2.4. This section describes the current state of ssNMRlib, containing some 140 different pulse sequences,
including a number of general homo- and heteronuclear correlation experiments, and a large panel of proton-, carbon- and
nitrogen-detected 1D, 2D, 3D and 4D experiments for resonance assignments, structure determination and measurements of
molecular dynamics. The nomenclature used below to describe experiments employs upper-case letters for nuclei of which the
chemical shift is edited and lower-case letters for nuclei which are not frequency-edited. The full list of experiments currently
in ssNMRlib is shown in Listing S2 in the Supplementary Information.

### 2.2.1 One- and two-dimensional experiments with $^{13}$C or $^{15}$N detection

$^{13}$C- and $^{15}$N-detected experiments are most useful for samples spinning at moderate MAS frequencies, typically using 3.2 mm
rotors (the use of 4 mm and 1.9 mm rotors can be implemented in a straightforward manner). ssNMRlib contains $^{13}$C-detected
and $^{15}$N-detected experiments with $^1$H-$^{13}$C/$^{15}$N CP or INEPT transfer or direct $^{13}$C/$^{15}$N excitation, as well as 1D versions of
double-CP hnCA and hnCO experiments. Experiments for measurement of $^1$H longitudinal relaxation (saturation recovery) is
useful for choosing an appropriate recycle delay; furthermore, $^{13}$C and $^{15}$N transverse (T$_2$') relaxation rate constants provide
a rapid view of the coherence life times, and may be useful for optimizing decoupling parameters.

Two-dimensional $^{13}$C-detected experiments comprise homo-nuclear $^{13}$C-$^{13}$C correlations with DARR (Takegoshi et al.,
2001), DREAM (Verel et al., 1998), ALFRESCO (Wi and Frydman, 2020), RFDR (Bennett et al., 1992) and CHHC (Lange





**Figure 5.** Optimization procedure of transfer steps. (A) Setup of BSH-CP CO-CA transfer optimization, found in Carbon detection – Calibrations – CC transfers. Several parameters (RF field, durations, trim pulses) can be optimized. The theoretical values, calculated for the frequency offset and MAS frequency are shown in the shell console, and in a popup window. The optimized acquisition parameters after optimization are automatically stored, and can be inspected by clicking on the cogwheel symbol (top windows). (B) Setup of $^1$H-$^{13}$C CP optimization. A choice of optimization parameters (RF field strengths, duration), different pre-defined ramps (e.g. linear 70-100 ramp) and carrier offsets are proposed.



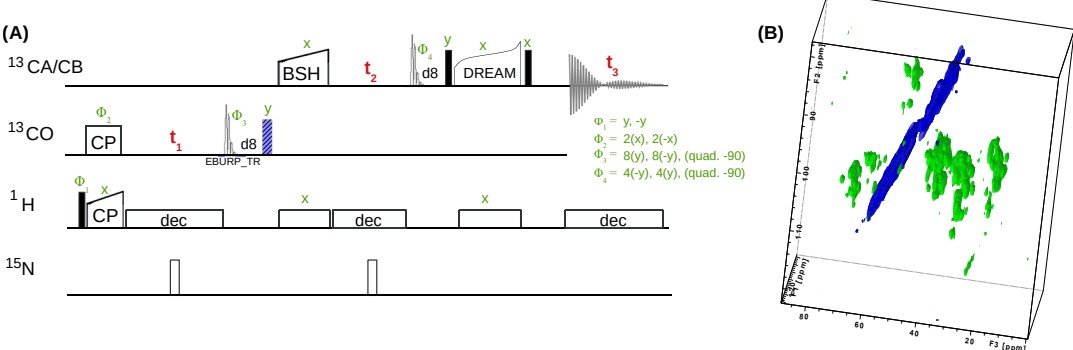

**Figure 6.** (A) 3D hCOCACB correlation experiment is composed of h-CO CP, CO-CA BSH-CP and CA-CB DREAM transfer steps. Filled and open bars denote $\pi/2$ and $\pi$ pulses, respectively. Unless otherwise noted, the phase is x. The $^{13}$C carrier frequency is changed during the pulse sequence as denoted by elements applied on either $^{13}$CO or $^{13}$CA/CB respectively. (B) Experiment done on a 600 MHz spectrometer with 15 kHz MAS, 25°C, with a $^{13}$C, $^{15}$N labeled sample of a protein assembly from bacteriophage Fraga et al. (2017).

et al., 2003) transfer. A $^{15}$N-detected proton-driven spin diffusion (PDSD) experiment and a NHHC experiment are also available. Additionally, $^{13}$C-$^{13}$C correlation experiments based on SPC5, S3 and various C- and R-sequences are under development. These pulse sequences are useful for a number of organic solids including biomolecules.

### 2.2.2 Carbon-detected resonance assignment experiments

Currently, a dozen $^{13}$C-detected experiments are present in ssNMRlib for high-dimensional (3D, 4D) correlation spectra. The majority of the presently available experiments are intended for protein resonance assignment, including 3D hNCACB, hNCACX, hNCOCX, hCANCO, hCONcaCB, hCOCACB and 4D hCANCOCX, hCONCACO and hCONCACB experiments, which use cross-polarization for H-X and N-C transfers. For $^{13}$C-$^{13}$C transfers, we have implemented several options, including DREAM (Verel et al., 2001), DARR (Takegoshi et al., 2001), RFDR (Bennett et al., 1992) and BSH-CP (Chevelkov et al., 2013) transfers. Figure 6 shows one backbone pulse sequence from the library, a 3D h-CO-CA-CB experiment with BSH-CP and DREAM transfer steps, which, to our knowledge, has not been proposed before, and its application to a 50 kDa protein that assembles to tube-like structures (Fraga et al., 2017).

### 2.2.3 Carbon-detected experiments for flexible systems

A suite of experiments has been implemented for application to flexible molecular systems, where CP-based transfers are inefficient, and scalar-coupling based transfer is the better choice for correlation spectroscopy. This kind of experiments has been successfully applied, e.g. to bacterial peptidoglycan cell walls (Kern et al., 2008) or flexible tails in proteins (Gao et al., 2013). We have implemented $^{1}$H-$^{13}$C and $^{1}$H-$^{15}$N HETCOR experiments, hNCA, hNCO, hNcoCA, hNcaCO and hCC correlation





spectra, all based on INEPT transfer steps between H-N, H-C, C-C and N-C nuclei. An additional hCC experiment with an

initial H-C CP transfer followed by a C-C INEPT transfer is particularly useful for systems with some degree of flexibility, that still have sufficiently large dipolar H-C couplings to make the initial H-C CP the most efficient choice. We have furthermore experiments with direct [13]C excitation followed by C-C INEPT transfer.

### 2.2.4   Proton-detected protein resonance assignment experiments

The largest part of the current ssNMRlib implementation comprises [1]H-detected experiments, which are ideally used in com-

bination with deuteration and reprotonation of e.g. amide, methyl or aromatic sites (Barbet-Massin et al., 2013; Fricke et al., 2017; Fraga et al., 2017; Gauto et al., 2019; Xiang et al., 2015; Linser et al., 2010; Zhou et al., 2007). When used at MAS frequencies above 60-100 kHz, fully protonated systems can also yield comparably good [1]H line widths (Stanek et al., 2016). The 2D experiments comprise basic [1]H-[15]N and [1]H-[13]C correlations and simultaneous [1]H-[15]N *and* [1]H-[13]C 2D correlation experiments, with either INEPT or CP transfers.

ssNMRlib currently comprises more than 40 different experiments for protein resonance assignment with [1]H$^N$ or [1]H$^\alpha$ detection (listed in Listing S2 in the Supplementary Information). The 3D and 4D experiments for protein resonance assignment comprise all possible correlation spectra of the amide [1]H-[15]N moiety with the [13]C$\alpha$, [13]C$\beta$, [13]C' and [15]N nuclei at both sides of the amide. 3D and 4D variants are implemented.

We have used the following general principles when designing these experiments:

- Cross-polarization is used for all transfers between [1]H and either [15]N or [13]C, as well as for the transfers between [15]N and [13]C (either direction). Additionally, for the case of the intra-residue $H_i$-$N_i$-$CA_i$ and the inter-residue $H_i$-$N_i$-$CA_{i-1}$ experiments, the variants with N-C TEDOR transfers are implemented.

- For C$\alpha$–C' or C'–C$\alpha$ transfers, we have implemented different coherence transfer elements: (i) refocused INEPT for

unidirectional experiments such as hCACONH or INEPT for out-and-back transfers such as in hcoCAcoNH; (ii) additionally, experiments with band-selective C'–C$\alpha$/C$\alpha$–C' cross-polarization (BSH-CP) have been implemented; this transfer, based on dipolar couplings, may outcompete INEPT-based experiments when fast [13]C coherence decay hampers the latter.

- For C$\alpha$–C$\beta$ transfers, we have implemented experiments based on scalar-coupling transfer, akin to solution-state HN-

CACB or HNcoCACB experiments; the setting of the transfer delay leaves the user the freedom to choose either full transfer to C$\beta$ or only partial transfer, which results in both C$\alpha$ and C$\beta$ signals.

- In experiments with INEPT-based [13]C-[13]C transfer and chemical-shift editing of one of the two involved carbons or both of them, the chemical-shift evolution is done in a constant-time (CT) manner. CT editing comes at no cost in terms of sensitivity, i.e. the FID along this dimension does not decay. CT editing requires that the maximum evolution time is





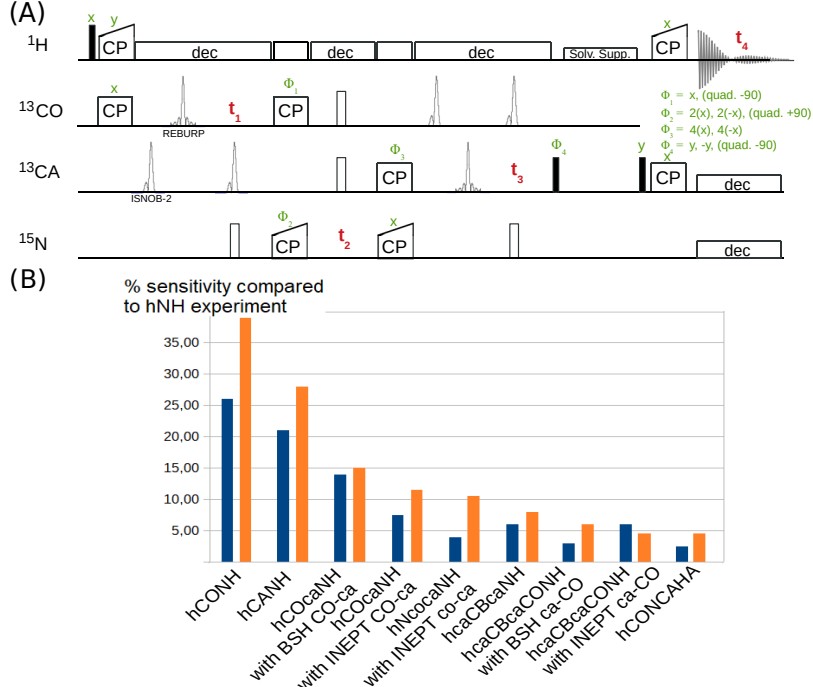

**Figure 7.** (A) 4D hCONCAHA pulse sequence for Hα-detected backbone correlation, using cross-polarization transfer steps. Filled and open bars denote $\pi/2$ and $\pi$ pulses, respectively. Homonuclear $^{13}$C-$^{13}$C decoupling is achieved with band-selective ISNOB2 pulses; Bloch-Siegert shift effects are compensated. Solvent suppression follows the ideas of the MISSISSIPPI scheme (Zhou and Rienstra, 2008), but unlike the original proposition uses a composite-pulse-decoupling scheme for saturating the solvent signals while storing the magnetization on the heteronucleus (+$C_z$). Quadrature detection is achieved with phases $\Phi_1$, $\Phi_2$ and $\Phi_4$, whereby the phase is changed by either +90 or -90 degrees, as indicated. (B) Sensitivity comparison of diverse pulse programs with a tube-like protein assembly formed by 50 kDa-large subunits (Fraga et al., 2017) and decameric assembly of 10x21 kDa, in orange and blue respectively. The data were obtained by integration of one-dimensional variants of the pulse sequences.

kept within the INEPT transfer delay, which is typically of the order of 6-7 ms. Additional semi-constant-time versions, which do not have this limitations are also implemented.

– In experiments that involve a dipolar-coupling based $^{13}$C-$^{13}$C transfer, e.g. with BSH-CP, no constant-time editing is done, because unlike the INEPT case above, the chemical shift cannot be edited during the transfer.

– We have systematically implemented homo-nuclear ($^{13}$C) decoupling in all indirect $^{13}$C dimensions, in particular CO
decoupling during CA frequency editing (and *vice versa*), using a band-selective inversion pulse (ISNOB-2, (Kupce et al., 1995)). This shape is advantageous because of its good compromise of pulse duration and cleanliness of the inversion profile. To correct the Bloch-Siegert shift, a second ISNOB-2 is applied to the decoupled band, and a REBURP refocusing pulse to the nucleus which is monitored. This implementation is illustrated in Figure 7A.





– In all indirect dimensions, heteronuclear decoupling is applied; when monitoring $^{13}$C frequencies, a $^{15}$N $\pi$ pulse is applied, and *vice versa*. $^{1}$H is decoupled with composite pulse decoupling (swfTPPM (Thakur et al., 2006) or WALTZ-16 (Shaka et al., 1983)). During $^{1}$H acquisition, composite-pulse decoupling is applied to $^{15}$N and/or $^{13}$C as appropriate (generally using the WALTZ-16 scheme (Shaka et al., 1983)).

As described above, all required band-selective decoupling pulses are calculated and set within the pulse sequence, and the correct shape(s) are called by ssNMRlib's python setup routines. The decoupling field strengths $\gamma \cdot B_1/(2\pi)$ can be specified in kHz by the user, and the setup script proposes values and sets the duration of the unit element pulse accordingly.

– For the majority of the assignment experiments, we have versions with additional deuterium decoupling. These experiments may be used with probes equipped for a $^2$H channel. The deuterium decoupling may increase coherence life times of carbons in deuterated proteins (see below).

Figure 7A shows one example pulse sequence, a four-dimensional hCONCAHA experiment with detection of the H$\alpha$ protons, which, to our knowledge, has not been published elsewhere. Figure 7B shows exemplary sensitivity comparisons of different experiments, for two different protein assemblies. The data are in good qualitative agreement with reported sensitivity comparisons presented elsewhere (Barbet-Massin et al., 2013; Fraga et al., 2017). It is interesting to compare e.g. INEPT vs BSH-CP transfer variants in experiments involving CO-CA transfers. The availability of pulse sequences with different transfer types allows the user to evaluate which experiment provides better sensitivity for a particular sample, before launching the 3D or 4D versions of these experiments.

### 2.2.5 Distance measurements for structure determination

Proton-detected experiments that probe $^{1}$H-$^{1}$H distances have been successful for determining structures of deuterated, partially reprotonated samples (Zhou et al., 2007; Linser et al., 2011; Huber et al., 2011; Knight et al., 2012). When combined with $^{13}$C and/or $^{15}$N dimensions, such 3D or 4D experiments provide highly useful structural information. These experiments have been successful even for determining structures of very large proteins, up to 12x 39 kDa (Gauto et al., 2019), and even fully protonated proteins (Agarwal et al., 2014; Andreas et al., 2016). Recoupling the $^{1}$H-$^{1}$H dipolar interaction has been achieved primarily with RFDR (Zhou et al., 2007; Linser et al., 2011; Knight et al., 2012), DREAM (Huber et al., 2011; Agarwal et al., 2014) and rotating-frame spin diffusion (Wittmann et al., 2016; Jain et al., 2017). It has been shown that $^{1}$H-$^{1}$H distances can be measured simultaneously for protons bound to $^{13}$C or $^{15}$N using simultaneous $^{1}$H-$^{13}$C and $^{1}$H-$^{15}$N CP steps, yielding simultaneously the connections between H$^{N}$-H$^{N}$, H$^{N}$-H$^{C}$ and H$^{C}$-H$^{C}$ distances (Linser et al., 2011; Gauto et al., 2019).

We have implemented RFDR and band-selective spectral spin diffusion (BASS-SD; Jain et al. (2017)) experiments, with versions yielding 3D experiments with either one or two $^{1}$H dimensions and either two or one $^{13}$C/$^{15}$N dimensions.



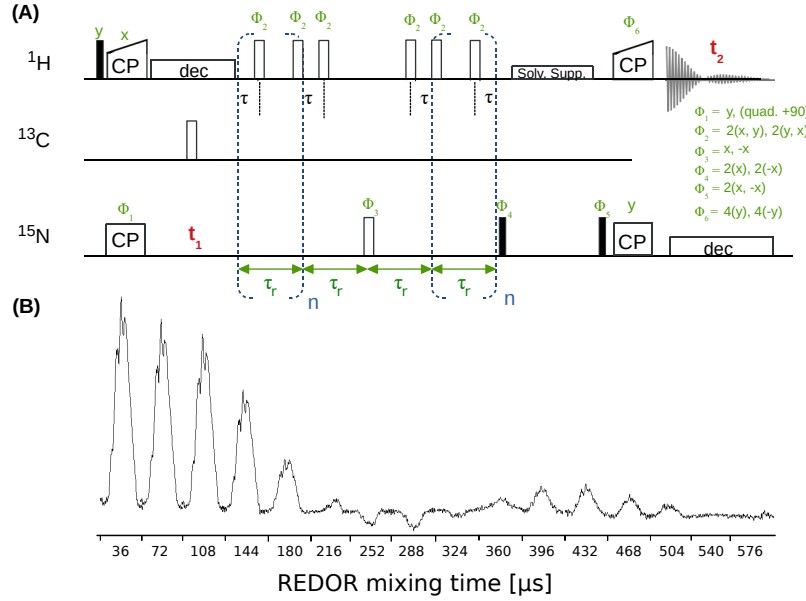

**Figure 8.** (A) hNH REDOR pulse sequence for measuring dipolar couplings, as implemented in ssNMRlib for $^1$H-$^{15}$N (shown here, reported in (Haller and Schanda, 2013)) and $^1$H-$^{13}$C dipolar couplings. Filled and open bars denote $\pi/2$ and $\pi$ pulses, respectively. The experiment has the possibility to shift the central $\pi$ pulse of each rotor period on the $^1$H channel by setting the delay $\tau$ (d7), which scales down the dipolar-coupling evolution, and therefore allows sampling the REDOR curve with more points and thus higher precision (Gullion and Schaefer, 1988; Schanda et al., 2010). (B) Experimental series of one-dimensional $^1$H-$^{15}$N REDOR spectra, collected at 55.555 kHz MAS, $^1$H and $^{15}$N $\pi$ pulses of 5 and 10 $\mu$s, respectively and a delay d7 of 3 $\mu$s (i.e. 0.5 $\mu$s between the closest consecutive $\pi$ pulses).

### 2.2.6 Experiments probing molecular dynamics

Solid-state NMR is ideally suited to investigate internal molecular motions, without the limitations that arise in solution-state NMR due to the overall molecular tumbling (Krushelnitsky and Reichert, 2005; Schanda and Ernst, 2016; Lamley and Lewandowski, 2016; Rovó, 2020). ssNMRlib contains a number of experiments that measure (i) dipolar couplings, which directly report on the order parameters of bonds, and (ii) longitudinal and transverse spin-relaxation parameters of $^{13}$C and $^{15}$N, which are sensitive to amplitudes and time scales of motion. (iii) Slower exchange dynamics can be probed by $^{15}$N or

$^{13}$C-edited exchange spectroscopy (EXSY, (Meier and Ernst, 1979)) or a version with simultaneous $^{15}$N and $^{13}$C editing, which are implemented in ssNMRlib. We furthermore have implemented chemical-exchange saturation transfer (CEST, Vallurupalli et al. (2012); Rovó and Linser (2018)) experiments to probe slow motions.

The dipolar-coupling measurements are based on a time-shifted REDOR experiment (Haller and Schanda, 2013; Schanda

et al., 2010), and are available with either CP or refocused-INEPT transfers. Figure 8 shows a hNH CP REDOR pulse sequence



and a series of one-dimensional REDOR experiments. Both $R_1$ and $R_{1\rho}$ relaxation experiments are available for $^{15}$N with either 2D (hNH) or 3D (hCONH) readout, and $^{13}$CO (with 3D hCONH readout) and $^{13}$C (with 2D hCH readout).

Additionally, $^1$H relaxation parameters can be measured with either $^1$H or $^{13}$C-detected pulse sequences implemented in ssNMRlib (see also section 2.2.1). $^1$H relaxation is generally challenging to interpret quantitatively (Schanda and Ernst, 2016), but the knowledge of longitudinal relaxation is useful for choosing the recycle delay that yields the highest signal-to-noise ratio per unit measurement time. Furthermore knowing the apparent $^1$H coherence life time is useful for setting up e.g. the maximum evolution time in indirect $^1$H dimensions.

## 2.3 Probe security handling in ssNMRlib

Ensuring that the power deposited in the probe is tolerated by the hardware is an important part of any NMR data acquisition. Topspin has a *Power-Check* feature, which verifies the RF peak power sent to the probe. If *Power-Check* is enabled, it does not allow acquisition if the power exceeds the limit on any channel. However, when applied for too long time even an RF field at lower power would destroy the probe, such that the built-in *Power-Check* is insufficient for avoiding hardware damage. Limiting the power to strictly exclude any possible probe damage is difficult, because hardware damage may arise as a complex function of deposited RF power, its duration and the duty cycle.

We have implemented power checks in ssNMRlib which verify peak power levels and durations of pulse sequence elements, e.g. CP elements, decoupling or hard pulses. The safety check is performed automatically by ssNMRlib whenever the user clicks on a button to load an experiment. In addition, a "Security" button in Topspin, right next to the button to launch NMRlib, as well as a button in the ssNMRlib window (see Figure 2), can be clicked at any time to control the user modified parameters.

The safety checks in ssNMRlib access a table (which can be modified by the user) that lists the accepted durations and power levels (in kHz) for a given element. This table comes with ssNMRlib. In the linux console from which Topspin was started, all the parameters that have been verified are listed, including whether they are within the limits of the probe (Figure 9). In case a parameter exceeds the specified power limits, a popup window warns the user, and the warning is written in the linux terminal. The user can nonetheless start the experiment by typing *zg*, despite a security warning. We have made this choice as we think that responsible NMR users, once warned, can take informed decisions. In the calibration experiments, which are usually launched automatically, a security check is performed and if it fails the acquisition is not started. The safety checks also verify the maximum power reached in RF ramps and parameter optimization (*popt*) arrays. The safety check is probe specific, i.e. a separate table of tolerated RF parameters is defined for each probehead.

The safety checks are currently bound to the naming convention and definitions used in a given experiment, as it will check, for example, the power level of the $^1$H RF field for a H-N CP (*cnst42*). When a user adds new experiments (see below), care should be taken to keep this naming convention, or to modify the security checks accordingly. The safety checks in ssNMRlib cannot exclude damage, but serves as a useful guide.





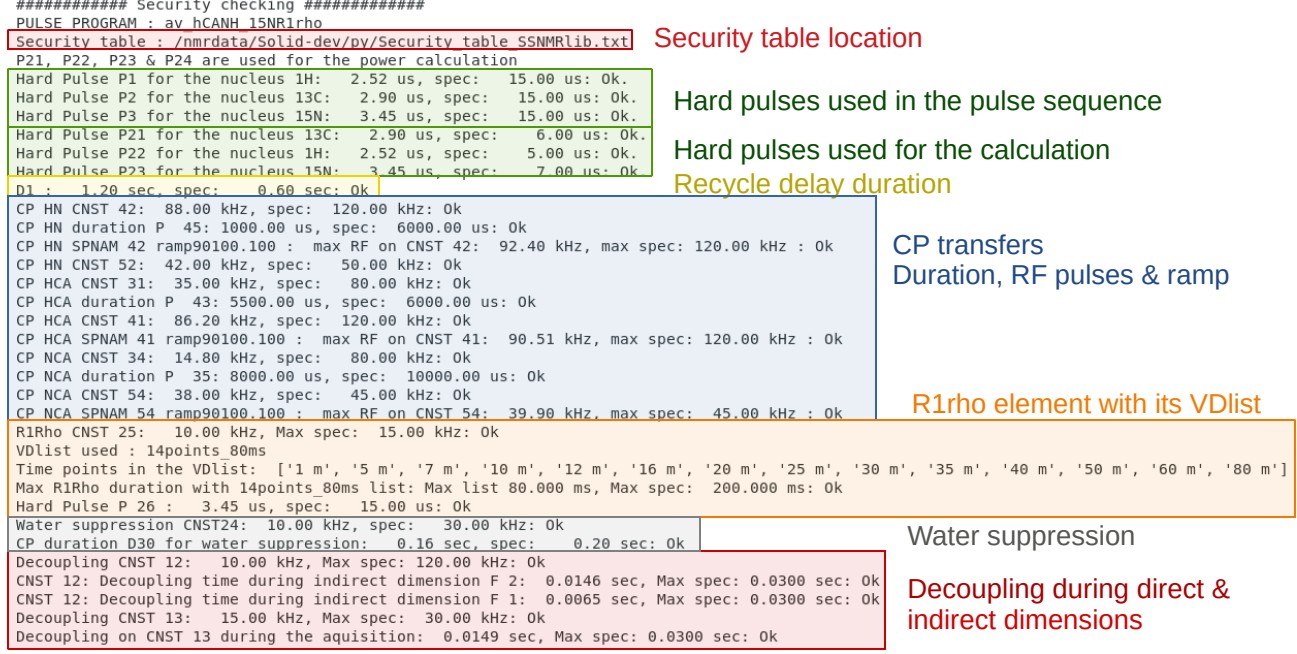

**Figure 9.** Example of security checks for the hCANH R1rho experiment. Shown is the output that ssNMRlib writes to the Linux console from which Topspin was started. All parameters used by the experiment are compared to values specified in a probe-specific table, and security warnings are issued in this console window as well as a popup window, in case a parameter exceeds the specifications.

## 2.4 Adding new experiments to ssNMRlib

ssNMRlib allows integrating an experiment into the library. Once integrated, the experiment is available as a button in the GUI window, and hitting this button will launch a script that sets up the experiment (pulse sequence, calculation or setting of acquisition parameters, setting of processing parameters etc, as described for the other experiments above). Integrating an experiment into ssNMRlib can be done by clicking the floppy disk icon in the GUI window. A popup window allows to specify the important parameters, and write a short manual for the experiment, which will appear when clicking the new button (S4 in the Supplementary Information). The pulse program, one jython script for acquisition and one for processing parameters are automatically generated. Additional functionalities could be added inside the script Jython as decoupling calculation, recall of constants or automatically security checks. By convention, we have set in the library:

- experiment.py : jython script containing experiment specific acquisition parameters.

- p_experiment.py : jython script defining experiment specific processing parameters.

- experiment_p.py : cogwheel symbol jython script used for non standard and experiment specific data analysis.

- experiment_sec.py : experiment specific security jython script.





When programming new pulse sequences, we strongly advice to use the naming convention for pulses, shapes, decoupling, delays etc., which is listed in Table S1.

## 2.5 Additional useful macros for processing and experiment setup

NMRlib comes with a number of additional useful macros and scripts for various tasks, including saving and reading acquisition
parameters, processing (e.g. phasing 3D or 4D experiments or summing spectra or time-domain data) or plotting (in Matplotlib format). These tools are described in the following sections.

### 2.5.1 Store and retrieve acquisition parameters and pulse sequences

ssNMRlib contains many useful tools for optimizing parameters, keeping the parameters in memory, and using the optimized parameters of all the required pulse sequence blocks when setting up a complex experiment. ssNMRlib allows writing all
the present parameters in a user-friendly text file, the "recap file". The file saves all currently stored parameters, along with the date/time of their creation, which allows the user to see which parameters have been recently optimized. We have also implemented the possibility to remove all optimized parameters, thus starting "from scratch" and avoiding the use of old parameters.

The recap file contains, in addition to the optimized RF parameters also information about the identity of the installed probe,
the cooling gas flow and temperature, MAS frequency, the location where the NMR data are stored.

In our experience, the recap file is a very useful tool to keep track and retrieve all experimental parameters after the experiments have been concluded, in a format that is more convenient than the Bruker acquisition file (*acqus*).

An example recap file is shown in Listing S3 in the Supplementary Information.

The same button also allows to read a previously stored "recap file". In this way, optimizations saved during a previous NMR
session can be automatically put back into the library. The user will thus be able to load any experiment from the library with the previously optimized parameters.

A *"Save your pulse program"* button allows to save the current pulse program into the present acquisition directory. This button is very useful e.g. when programming/editing pulse sequences: the user can save the present state of a sequence along with the data. This allows to more easily develop or debug pulse sequences and retrieve a previous state of a sequence.

### 2.5.2 Processing macros

General processing tools are part of NMRlib and have been introduced before (Favier and Brutscher, 2019). As ssNMRlib is part of this library, all previously proposed tools are also available for solid-state NMR experiments. Tools are available to add two multi-dimensional spectra (in time- or frequency domain), to phase multi-dimensional (3D, 4D) spectra, or to remove (i.e. set to zero) corrupt FIDs from a multi-dimensional data set, which may arise for example as a consequence of arching in
ssNMR experiments using python package (numpy Van Der Walt et al. (2011), nmrglue Helmus and Jaroniec (2013), matplotlib Hunter (2007)). Another macro allows to correct FIDs for magnetic-field drift (i.e. shearing of the spectrum), which shall be

particularly useful for solid-state NMR experiments, where it is generally not possible to use a deuterium lock to stabilize the field. Other routines, described earlier (Favier and Brutscher, 2019) allow generating nmrPipe (Delaglio et al., 1995) processing scripts, or to perform a circular shift to correct for potential mis-settings of the carrier frequency.

## 3 Conclusions and perspectives

We have presented herein ssNMRlib, a comprehensive library of pulse sequences and setup scripts which markedly facilitate solid-state NMR data acquisition, security checks, tracking of parameters and data handling. We developed the library in a multi-spectrometer facility, and the fact that pulse sequences are all centralized contributes greatly to keeping data acquisition simple and reproducible. While the current ssNMRlib version has its focus on pulse programs for biomolecular applications, the library can be extended in a straightforward manner to any kind of NMR data acquisition.

We are currently implementing more experiments, such as C- and R-type (Levitt, 2002) sequences, as well as $^{31}$P and $^{19}$F experiments. We foresee in future versions of ssNMRlib the use of the *WaveMaker* library, which allows to create shape files (ramps, selective pulses, decoupling sequences) directly from within the pulse sequence on the fly. This would allow performing a parameter optimization for shape files (e.g. adiabaticity), which currently requires to change the shape file name manually.

We encourage our colleagues in the field to add their pulse sequences, and hope that the platform we created will be useful for a wide range of NMR applications. While a user may simply add her/his experiments to the own local library, it facilitates also the exchange of experiments between laboratories.

*Code availability.* The ssNMRlib package is available from the authors upon request and at the following URL:
http://www.ibs.fr/nmrlib

*Author contributions.* A. V. and P. S. wrote the pulse sequences contained in ssNMRlib and designed the workflow of ssNMRlib and the organization of the library. A. F. adapted the NMRlib software structure for the integration of ssNMRlib. B. B. contributed calculations for selective pulses and other pulse-programming snippets. A. V. and P. S. prepared figures. P. S. wrote the manuscript with input from all authors.

*Competing interests.* The authors declare that they have no competing interests.

*Acknowledgements.* This work used the NMR and isotope labeling platforms of the Grenoble Instruct center (ISBG; UMS 3518 CNRS-CEA-UJF-EMBL) with support from FRISBI (ANR-10-INSB-05-02) and GRAL (ANR-10-LABX-49-01) within the Grenoble Partnership for Structural Biology (PSB). P. S. acknowledges support from the European Research Council (ERC-StG-311318 - ProtDyn2Function). We



thank Johanna Becker-Baldus for testing ssNMRlib and many useful suggestions of additional features. We thank Audrey Hessel, Charles-Adrien Arnaud and Cecile Breyton for providing samples used for testing and setup of ssNMRlib, and Rasmus Linser, Sabine Hediger and

Daniel Lee for discussions about pulse sequences.




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
