# Peer review of "ssNMRlib: a comprehensive library and tool box for acquisition of solid-state NMR experiments on Bruker spectrometers"

_Magnetic Resonance, 2020_

## Referee Comment (RC1) · Anonymous Referee #1 · 12 Oct 2020

The manuscript **mr-2020-25**, **ssNMRlib: a comprehensive library and tool box for acquisition of solid-state NMR experiments on Bruker spectrometers** by *Vallet et al.* introduces a solid-state NMR pulse sequence library, accompanied by a setup tool specifically for Bruker spectrometers. Sadly, the NMR structural biology community is far behind other communities such as X-ray and EM in terms of automation. Therefore, the step towards more automation to save experimental time is in general commendable.

I have a few minor remarks:

1. The setup was tested on **AVIII** and **NEO** consoles. The authors should comment, if or how their setup and pulse sequences are compatible with **AVII** con-
soles. Despite the fact that the support for **AVII** consoles is declining, they are
still widespread within the NMR community.

2. How about compatibility on a spectrometer work station running **Microsoft Win-
dows** OS?

3. Obviously, frequency units are more intuitive in the NMR perspective. However,
from the technical point of view the use of power levels in Watts alerts the user
more than using frequency units, as the nutation by $X$ kHz can require very low
or very high power, depending on the $90°$ pulse length. Can the authors comment
on how their safety check traps this potential risk?

4. Along these lines, the authors implemented safety measures by checking for
overshooting RF power for specific pulse elements. However, RF limits are given
here in units of kHz, which might be risky as the absolute power integral is cru-
cial. The authors should add more details on how exactly the safety checks are
implemented as it is still unclear to the reader by which criteria the margins are
set.

5. The setup tool presented in this work does not provide much novelty as it is very
similar to the tools offered by the manufacturer, namely **TopSolids** and **bioTop**.
The latter is largely for solution-state experiments, but the functionalities, such as
creating experimental templates, is provided as well as an automated calibration
function, and, I believe, solid-state setups are being currently included. Further-
more, **TopSpin** already provides a large and growing number of solid-state pulse
sequences for biomacromolecules, but also for materials, which is not included
in the library compiled by the authors. In my opinion, it would be more helpful
to the NMR community to push the manufacturer to improve the tools that they
already have to the desires of the users and provide them with state-of-the-art
pulse sequences.

In conclusion, it is a nice attempt and the authors introduce some new assignment experiments, however, the setup is only a minor advancement compared to already existing tools. In my view, joining forces with Bruker would be much more fruitful in terms of an universal solution for the NMR community.

---

## Referee Comment (RC2) · Anonymous Referee #2 · 20 Oct 2020

The manuscript by Vallet et al introduces a library of pulse sequences for Bruker spectrometers that enables the users to easily implement a wide range of experiments commonly employed for resonance assignments or probing dynamics in solid state NMR.

While ssNMRlib is not a novel experimental technique, it could be of great use for users with different levels of expertise in setting up solid-state NMR experiments. The features that I find useful are:

(1) Easy optimisation protocol for coherence transfer steps that are the building blocks for complex resonance assignment experiments. (2) The ease with which a new experiment can be added to the library. (3) The ease with which the experimental parameters can be retrieved, especially RF powers in kHz units.

I would like to ask the following questions to the authors:

(1) The authors implement a security feature that adds one more layer to the "Power Check" feature of Topspin, which is concerned with duration. A full proof security feature is probably difficult to set up, but could the authors please comment about switching to a "duty cycle" based security system?

(2) As the authors promise, indeed it would be great if on-the fly shape generation could be implemented. It would also be helpful if optimal control derived shapes are available in ssNMRlib. However, judging from the setup of ssNMRlib, this can be achieved with reasonable efforts also from the users.

(3) Like any other software, it would be great to have a "Troubleshooting" section that would help the beginners to address problems. In this regard, also an email discussion group among users would be beneficial.

(4) It is well accepted among NMR users that Topspin is not really well suited for performing quantitative data analysis. By integrating ssNMRlib with programs like nmrglue and matplotlib, the users can extend the workflow from setting up experiments to even reliable data analysis. Do the authors already plan to add features like this in ssNMRlib?

To conclude, I find the manuscript interesting and publishable in Magnetic Resonance.
* * *

---

## Short Comment (SC1) · 21 Oct 2020

The authors present a tool box for simple setup of ssNMR experiments on Bruker spectrometers. ssNMRlib comprises a large variety of experiments both for calibration and for analysis of protein samples, e.g. for assignment, distance restraints or dynamics. In my opinion, ssNMRlib provides several benefits both for individual scientists and for the scientific community:

1) The library of already implemented experiments is valuable especially for beginners to get an idea, which experiments are available and might be used. A collection of pulse programmes is also useful for scientists who want to try new types of experiments, but

cannot write pulse programmes (yet) on their own.

2) Power levels in kHz are convenient to quickly check resonance conditions or duty cycles.

3) The recap files are helpful for reporting experimental parameters. Unfortunately, it still happens in papers that important NMR parameters are not reported or not reported in a meaningful manner. The recap files could (maybe in parts) be reported in supplementary information of research articles to facilitate assessment of results or reproducibility of experiments.

4) For the same reason, I support the author's invitation that other scientists should add their (newly developed) experiments to the library. Doing so has a chance to accelerate spreading new pulse programmes among different labs.

To conclude, I consider ssNMRlib a useful tool for training of new NMR spectroscopists, for documentation and for spreading new techniques in a standardised way.

---

## Short Comment (SC2) · 23 Oct 2020

I think that this work is a nice step forward for the community. I have worked with Bruker previously on the topsolids project, doing similar things to what is presented in the paper. I am happy to see another option that is, perhaps, a bit more flexible than the manufacturer's release. The power handling and safety checks are very much appreciated, and necessary to prevent bone-headed mistakes. In my experience, this approach would be back-compatible with AVII systems with just a little bit of work, if it doesn't already work, especially if the guts are mostly in python.

It also appears as though it would be relatively easy to retrofit into existing pulse se-

quences using the header equations, which would make switching between pulse programming "styles" fairly seamless. Just a bit of bookkeeping to set it up.

I also think that this framework can be adapted for use with materials experiments like MQMAS or an HMQC, although the book-keeping demands for those experiments is generally less severe than for biological experiments. The optimization protocols presented here would be a great thing for materials experiments, and should help with the collection of such experiments. The protocols and pre-calculations must be written, but that should be a small task for someone familiar with the experiments.
* * *

---

## Author Comment (AC1) · 12 Nov 2020

Thank you for the positive feedback.

Replying to: "It also appears as though it would be relatively easy to retrofit into existing pulse sequences using the header equations, which would make switching between pulse programming "styles" fairly seamless. Just a bit of bookkeeping to set it up.":

Our reply: We think that it should be rather straightforward to fit existing pulse sequences into the NMRlib format. To do it in a clean manner, the names of the parameters (CPs, pulse lengths, durations etc) should be the those specified in the namingconvention file given in the Supporting Information.

Replying to the possibility to include MQMAS and HMQC experiments:

Our reply: Yes, it would be great to extend NMRlib also to quadrupolar and materials applications. In principle the library is open to any experiment. Feel free to reach out if we can help with ideas. We are not experts in MQMAS applications.

———————————————

---

## Author Comment (AC2) · 12 Nov 2020

Good to have this positive feedback, thank you.

————————————————————

---

## Author Comment (AC3) · 12 Nov 2020

Replying to: "1. The setup was tested on AVIII and NEO consoles. The authors should comment, if or how their setup and pulse sequences are compatible with AVII consoles. Despite the fact that the support for AVII consoles is declining, they are still widespread within the NMR community."

Our reply: We have developed NMRlib on AVIII, and successfully tested it on NEO consoles. We do not think that it currently works on AVII, nor how much effort it would be to make it work on AVII. Note that Trent Franks writes in his comment (see SC2): "In my experience, this approach would be back-compatible with AVII systems with just

a little bit of work, if it doesn't already work, especially if the guts are mostly in python."

Replying to: "2. How about compatibility on a spectrometer work station running Microsoft Windows OS?"

Our reply: We only can give a similar response than to point 1. As our spectrometers are all equipped with LINUX workstations, we only have experience with this environment. Therefore, we recommend to install a Linux computer, to make NMRlib available on a spectrometer.

Replying to "3. Obviously, frequency units are more intuitive in the NMR perspective. However, from a technical point of view the use of power levels in Watts alerts the user more than using frequency units, as the nutation by X kHz can require very low or very high power, depending on the 90âŮę pulse length. Can the authors comment on how their safety check traps this potential risk? "

Our reply: We considered the options of checking either Watt or kHz. We finally opted for kHz for two reasons: (i) the probe specifications provided by the manufacturer are generally in kHz. One would find on such a specification sheet, e.g. the one of our 1.3 mm probe: "the probe can provide >170 kHz For pulse lengths up to 50 ms". How this translates to Watt depends on the setup. We, thus, thought that it is more general to provide a table with kHz power limitations. (ii) the existing "Power Check" feature that is by default enabled in Topspin checks already for Watt power levels. The peak power level that the Power Check verifies is probe-specific, and defined by the Bruker engineer during installation or hardware control. The check in kHz is an additional layer, on top of the Power Check. The kHz calculation depends on the correct calibration of the reference pulse lengths (p21, p22, p23). If those are incorrectly set, the calculation will go wrong, and NMRlib would calculate possible too high power levels (in Watt). We partly capture such a scenario by also doing safety checks on p21, p22, and p23. If those pulse lengths are odd, then the safety check alerts the user.

Replying to : "4. Along these lines, the authors implemented safety measures by

checking for overshooting RF power for specific pulse elements. However, RF limits are given here in units of kHz, which might be risky as the absolute power integral is crucial. The authors should add more details on how exactly the safety checks are implemented as it is still unclear to the reader by which criteria the margins are set. "

Our reply: The safety check feature goes through all the parameters of a given experiment and compares the values to those provided in a table. This table is listed in a file, and the user can check, edit, and chnage those values. The python script that does the safety check is experiment-specific, because the parameters used in a given experiment are different from one pulse sequence to the next. As stated above, the safety check, operating in kHz rather than Watt, would only have a problem if the reference pulses, from which power levels (e.g. for CP) are calculated, are wrong. For example, if the reference 90 degree pulse length p21 is mis-set 2 times too long, the calculation would indicate that at that power level (plw2) the obtained RF field is 2 times lower than what it really is. Accordingly, the calculation of a power level for a CP, which is based on this reference pulse, would have as a result a too high power (in Watt). This might possibly cause problems. We circumvent the problem by checking also whether the reference power levels are in a reasonable range. We do not claim that this safety check is 100% safe. Nonetheless, they turn out to remove a vast majority of problems. We are open to suggestions how to improve the safety check feature! We have now added an entire section to the updated Supporting Information, which explains in more detail how the safety checks work.

We have attached the new part of the Supporting Information to this message.

Replying to question 5., related to the implication of Bruker in this process:

Our reply: In principle we agree that it would be best if Bruker did this work, or if they integrated this library, and ensured its continuity over time. We have indeed discussed with Bruker, and while they are generally interested, it is a very slow and time-consuming process to get Bruker practically involved in such a development. As an
aside, currently Topsolids lacks largely the 1H-detected bio-ssNMR experiments, currently. While it is not clear how Bruker's tools will evolve, we think that it makes sense to just move forward and make this tool freely available to the academic community. In conclusion, it is a nice attempt and the authors introduce some new assignment experiments, however, the setup is only a minor advancement compared to already existing tools. In my view, joining forces with Bruker would be much more fruitful in terms of an universal solution for the NMR community.

We thank the reviewer for the overall positive assessment.

Please also note the supplement to this comment:
https://mr.copernicus.org/preprints/mr-2020-25/mr-2020-25-AC3-supplement.pdf

———————————————————

[Figure]

**Supplement:**

**Listing S5 : Security checks – How does it work?**

Each pulse sequence within the solid-state NMRlib module is associated with a jython script named as "experiment_p.py". This script allows the user to define selectively which parameters will be verified for this specific experiment.
In order to perform the safety checks, the safety script will compare : the parameter value within the topspin experiment with the safety table adapted by the user for his probes.
This safety table, so-called "Security_table_SSNMRlib.txt", is located in : "/NMRlib/py" and contains all the maximum values authorized for each probe.
An extract of this file is shown below.

| SIZE | PULSE_TYPE | ELEMENT | DEFINITION | CNST_NAME | MAX_kHz | PULSE | DURATION | UNIT | SPNAM | MIDPOINT_RAMP | SEQ_DETECTION |
|---|---|---|---|---|---|---|---|---|---|---|---|
| 0.7 | HP | 1H | 1H_HP | - | - | P1 | 15 | us | - | - | 1H |
| 0.7 | HP | 1H | 1H_HP | - | - | P2 | 15 | us | - | - | 13C |
| 0.7 | HP | 1H | 1H_HP | - | - | P2 | 15 | us | - | - | 15N |
| 0.7 | HP | 13C | 13C_HP | - | - | P1 | 15 | us | - | - | 13C |
| 0.7 | HP | 13C | 13C_HP | - | - | P2 | 15 | us | - | - | 1H |
| 0.7 | HP | 13C | 13C_HP | - | - | P3 | 15 | us | - | - | 15N |
| 0.7 | HP | 15N | 15N_HP | - | - | P1 | 15 | us | - | - | 15N |
| 0.7 | HP | 15N | 15N_HP | - | - | P3 | 15 | us | - | - | 13C |
| 0.7 | HP | 15N | 15N_HP | - | - | P3 | 15 | us | - | - | 1H |
| 0.7 | HP | 2H | 2H_HP | - | - | P4 | 25 | us | - | - | 1H |
| 0.7 | HP | 2H | 2H_HP | - | - | P4 | 25 | us | - | - | 13C |
| 0.7 | HP | 2H | 2H_HP | - | - | P4 | 25 | us | - | - | 15N |
| 0.7 | HP | 13C | 13C_HP_for_CP | - | - | P21 | 5 | us | - | - | 1H |
| 0.7 | HP | 13C | 13C_HP_for_CP | - | - | P21 | 5 | us | - | - | 13C |
| 0.7 | HP | 13C | 13C_HP_for_CP | - | - | P21 | 5 | us | - | - | 15N |
| 0.7 | HP | 1H | 1H_HP_for_CP | - | - | P22 | 4 | us | - | - | 1H |
| 0.7 | HP | 1H | 1H_HP_for_CP | - | - | P22 | 4 | us | - | - | 13C |
| 0.7 | HP | 1H | 1H_HP_for_CP | - | - | P22 | 4 | us | - | - | 15N |
| 0.7 | HP | 15N | 15N_HP_for_CP | - | - | P23 | 6 | us | - | - | 1H |
| 0.7 | HP | 15N | 15N_HP_for_CP | - | - | P23 | 6 | us | - | - | 13C |
| 0.7 | HP | 15N | 15N_HP_for_CP | - | - | P23 | 6 | us | - | - | 15N |
| 0.7 | HP | 2H | 2H_HP_for_CP | - | - | P24 | 25 | us | - | - | 1H |
| 0.7 | HP | 2H | 2H_HP_for_CP | - | - | P24 | 25 | us | - | - | 13C |
| 0.7 | HP | 2H | 2H_HP_for_CP | - | - | P24 | 25 | us | - | - | 15N |
| 0.7 | CP | HN | RF_Field_1H | CNST42 | 190 | P45 | 6000 | us | 42 | 62 | - |
| 0.7 | CP | HN | RF_Field_15N | CNST52 | 55 | P45 | 6000 | us | - | - | - |
| 0.7 | CP | HC | RF_Field_13C | CNST31 | 70 | P43 | 6000 | us | - | - | - |
| 0.7 | CP | HC | RF_Field_1H | CNST41 | 190 | P43 | 6000 | us | 41 | 61 | - |
| 0.7 | CP | HCA | RF_Field_13C | CNST31 | 70 | P43 | 6000 | us | - | - | - |
| 0.7 | CP | HCA | RF_Field_1H | CNST41 | 190 | P43 | 6000 | us | 41 | 61 | - |
| 0.7 | CP | HCO | RF_Field_13C | CNST31 | 70 | P43 | 6000 | us | - | - | - |
| 0.7 | CP | HCO | RF_Field_1H | CNST41 | 190 | P43 | 6000 | us | 41 | 61 | - |
| 0.7 | CP | HACA | RF_Field_13CA | CNST35 | 70 | P34 | 7000 | us | - | - | - |
| 0.7 | CP | HACA | RF_Field_1HA | CNST45 | 190 | P34 | 7000 | us | 45 | 57 | - |
| 0.7 | CP | NCO | RF_Field_13CO | CNST33 | 70 | P53 | 10000 | us | - | - | - |
| 0.7 | CP | NCO | RF_Field_15N | CNST53 | 55 | P53 | 10000 | us | 53 | 63 | - |
| 0.7 | CP | NCA | RF_Field_13CA | CNST34 | 70 | P35 | 10000 | us | - | - | - |
| 0.7 | CP | NCA | RF_Field_15N | CNST54 | 55 | P35 | 10000 | us | 54 | 60 | - |
| 0.7 | CP | sim_HCN | RF_Field_13C | CNST31 | 70 | P43 | 6000 | us | - | - | - |
| 0.7 | CP | sim_HCN | RF_Field_1H | CNST41 | 190 | P43 | 6000 | us | 41 | 61 | - |
| 0.7 | CP | sim_HCN | RF_Field_15N | CNST51 | 55 | P43 | 6000 | us | - | - | - |

The security check is automatically performed when an experiment is loaded from NMRlib. Moreover, if the user wants to change a parameter inside topspin, a "security button" can be hit, at any time, in order to re-do all the checks.
If a problem appears, a pop-up is generated showing the actual problematic topspin value and the specification in order to alert the user. This warning is also written in the topspin terminal.

```
############ Security checking #############
PULSE PROGRAM : av_hNH_cp_cp_miss.IBS
Security table : /home/avallet/NMRlib/py/Security_table_SSNMRlib.txt
P21, P22, P23 & P24 are used for the power calculation
Hard Pulse P1 for the nucleus 1H:   2.75 us, spec:   15.00 us: Ok.
Hard Pulse P2 for the nucleus 13C:  2.93 us, spec:   15.00 us: Ok.
Hard Pulse P3 for the nucleus 15N:  3.80 us, spec:   15.00 us: Ok.
Hard Pulse P21 for the nucleus 13C:  2.93 us, spec:   6.00 us: Ok.
Hard Pulse P22 for the nucleus 1H:  2.75 us, spec:   5.00 us: Ok.
Hard Pulse P23 for the nucleus 15N:  3.80 us, spec:   7.00 us: Ok.
D1 :   0.89 sec, spec:    0.60 sec: Ok
CP HN CNST 42:  15.00 kHz, spec:  120.00 kHz: Ok
CP HN duration P  45: 1000.00 us, spec:  6000.00 us: Ok
CP HN SPNAM 42 ramp50100.100 :  max RF on CNST 42:  18.75 kHz, max spec: 120.00 kHz : Ok
WARNING CP HN CNST 52:  51.00 kHz, spec:    50.00 kHz
Water suppression CNST24:  10.00 kHz, spec:   30.00 kHz: Ok
CP duration D30 for water suppression:   0.16 sec, spec:    0.20 sec: Ok
Decoupling CNST 12:   10.00 kHz, Max spec: 120.00 kHz: Ok
CNST 12: Decoupling time during indirect dimension F 1:  0.0263 sec, Max spec: 0.0300 sec: Ok
Decoupling CNST 13:    5.00 kHz, Max spec:  30.00 kHz: Ok
Decoupling on CNST 13 during the aquisition:  0.0150 sec, Max spec: 0.0300 sec: Ok
```

Security Warning ✕

CP WARNING: HN out of specification

You will send too much power to the probe!
Actual CNST 52: 51.00 kHz, Max. spec: 50.00 kHz

Close

In this way, each pulse sequence has its own personalized security which is adapted for its use and to the probe specification :

For each pulse sequence :
- All the parameters (water suppression elements, pulse, duration & RF power) within the pulse sequence are checked according to the probe.
- In a ramped RF shape, for CP or DREAM/BSH, the maximum value of the ramp is taken into account. To this end, the maximum value in the shape file is extracted, and the corresponding kHz value is calculated.
- The safety check verifies that midpoint of the ramp that is stored in a constant (e.g. cnst62 = 95 for a 90-to-100 ramp) and used for calculating the power level of the ramp, is indeed the mean of that ramp. This is done by explicitly verifying the ramp in the shape file.
- In CP experiments, the values of the two corresponding RF fields is checked. If the values are too far from Hartmann-Hahn conditions a popup window informs the user. This may point to mis-calibration of the reference pulses, as the CP power levels are calculated from them.
- If a list is used in the pulse sequence (e.g. the spin-lock duration in an $R_{1\rho}$ experiment), the safety check retrieves all the values within this list and determines the maximal value. This ensures that all values are within the safety limits.
- If an element is repeated multiple times, the safety check will verify the total duration, i.e. the duration of the unit element times the number of repeats.
- The decoupling times are retrieved automatically for the whole direct and indirect dimensions, i.e. the maximum evolution time ($t_{1max}$) is considered.

In addition, for the calibrations :
- As the parameter optimization array is define within the python script, negative values are automatically discarded.
- In a parameter optimization, the safety check verifies that none of the experiments of the array exceeds the allowed safety limits.
- Clicking the button of a calibration experiment generally starts the acquisition automatically. However, if a safety problem is detected, the calibration is not launched automatically.

---

## Author Comment (AC4) · 12 Nov 2020

Replying to question (1), about the safety check and duty cycle:

Our reply: There a multiple ways, how one could implement security checks for a given pulse sequence, and all have their pros and cons. In the present version of ssNMRlib, we have opted for checking a range of pulse sequence parameters by comparing them to values listed in a file (see my reply to reviewer 1). In the future, we may also exploit the possibility of implement security features based on duty cycle calculations.

Replying to question (2), about on-the fly shape generation:

Our reply: WaveMaker is part of the NMRlib distribution, and it has been used for a number of solution state NMR pulse sequences. Actually, when a new data set is loaded, WaveMaker is automatically executed, which makes it easy to use (or not) wavemaker commands in new pulse sequences.

For ssNMRlib, we have considered using WaveMaker to create shapes. WaveMaker would have one advantage over the current implementation: it would allow to calculate shapes on the fly, including arbitrary shapes, e.g. tangential CP ramps. WaveMaker would allow doing e.g. a popt optimization of the shape file directly within one popt run, rather than by comparing different shapes in different experiments.

We ended up deciding against implementing WaveMaker at the current stage. One reason is that we would have to go back and change some 140 pulse sequences; we decided for now not to do it and rather have a functional library in place. The second reason is that we wanted to have a coherent naming convention, where each type of CP (e.g. H-N, H-C, etc) would have the same name across all experiments. This means that we have almost all of the available constant names (cnst) assigned to CPs or other parameters. If we wanted to use WaveMaker, we would need to have additional parameters, such as the adiabaticity. The number of constants in Topspin is limited to 64 (cnst0 to cnst63). We cannot fit all constants that we would need within these 64. This is not a good reason, and we hope that Topspin will alleviate this limitation soon.

Replying to question (3) about a "Troubleshooting" section and discussion group:

Our reply: This is a good idea. We have thought of a forum already, and will set it up soon.

Replying to question (4), related to the use of python/matplotlib/nmrglue:

Our reply: A number of macros in Topspin already use python scripts for data analysis (including nmrglue). For example, there are scripts in NMRlib which fit diffusion or overall-tumbling (TRACT) data. Likewise, scripts are available to export spectra as
matplotlib figures.

These scripts are executed e.g. by clicking the cogwheel symbol right next to the DOSY or TRACT setup buttons in the solution-state library. Those examples indicate how to extract and export data to python/matplotlib. The software is completely open to exporting data using python, in the way that is indicated with these examples.

We thank the reviewer for the positive evaluation.
* * *

---

## Author Response (AR1)

The manuscript **mr-2020-25**, **ssNMRlib: a comprehensive library and tool box for acquisition of solid-state NMR experiments on Bruker spectrometers** by *Vallet et al.* introduces a solid-state NMR pulse sequence library, accompanied by a setup tool specifically for Bruker spectrometers. Sadly, the NMR structural biology community is far behind other communities such as X-ray and EM in terms of automation. Therefore, the step towards more automation to save experimental time is in general commend- able.

I have a few minor remarks:

1. The setup was tested on **AVIII** and **NEO** consoles. The authors should comment, if or how their setup and pulse sequences are compatible with **AVII** consoles. Despite the fact that the support for **AVII** consoles is declining, they are still widespread within the NMR community.

We have developed NMRlib on AVIII, and successfully tested it on NEO consoles. We do not think that it currently works on AVII, nor how much effort it would be to make it work on AVII.

We have added explicitly that it has been tested on AVIII and NEO. In the section "Implementation of ssNMRlib" the sentence now reads:

*"NMRlib is an add-on in the Bruker Topspin software, and has been tested currently on Topspin 3.2 to 3.6 and Topspin 4, on Linux host computers, and AVANCE III and NEO consoles."*

We do not want to write explicitly that AVII is not possible because it may work. We cannot test it.

2. How about compatibility on a spectrometer work station running **Microsoft Windows** OS?

We only can give a similar response than to point 1. As our spectrometers are all equipped with LINUX workstations, we only have experience with this environment. Therefore, we recommend to install a Linux computer, to make NMRlib available on a spectrometer.

See point 1. above for the statement where we explicitly say that it works with Linux. There may be a workaround for Windows, but we do not know.

3. Obviously, frequency units are more intuitive in the NMR perspective. However, from a technical point of view the use of power levels in Watts alerts the user more than using frequency units, as the nutation by X kHz can require very low or very high power, depending on the $90°$ pulse length. Can the authors comment on how their safety check traps this potential risk?

We considered the options of checking either Watt or kHz. We finally opted for kHz for two reasons:

(i) the probe specifications provided by the manufacturer are generally in kHz. One would find on such a specification sheet, e.g. the one of our 1.3 mm probe: "the probe can provide >170 kHz

For pulse lengths up to 50 ms". How this translates to Watt depends on the setup. We, thus, thought that it is more general to provide a table with kHz power limitations.

(ii) the existing "Power Check" feature that is by default enabled in Topspin checks already for Watt power levels. The peak power level that the Power Check verifies is probe-specific, and defined by the Bruker engineer during installation or hardware control. The check in kHz is an additional layer, on top of the Power Check.

This being said, we agree that an additional check in Watt could be useful. In particular, the kHz calculation depends on the correct calibration of the reference pulse lengths (p21, p22, p23). If those are incorrectly set, the calculation will go wrong, and NMRlib would calculate possible too high power levels (in Watt).

We partly capture such a scenario by also doing safety checks on p21, p22, and p23. If those pulse lengths are odd, then the safety check alerts the user.

In the section "Probe security handling in ssNMRlib", we have added the following sentences:

*"We have decided to check RF power levels in kHz rather than in Watt, because probe specifications generally given by the manufacturer are in kHz; most spectroscopists also have a more intuitive understanding of what a probe can sustain, in kHz, rather than in Watt."*

Furthermore, we have added a detailed description to the Supplementary information.

4.          Along these lines, the authors implemented safety measures by checking for overshooting RF power for specific pulse elements. However, RF limits are given here in units of kHz, which might be risky as the absolute power integral is crucial. The authors should add more details on how exactly the safety checks are implemented as it is still unclear to the reader by which criteria the margins are set.

We have now added an entire section to the updated Supporting Information, which explains in more detail how the safety checks work.

The sentence we added is: *"A more detailed description of the safety check is provided in the Supporting Information (Listing 5)."*

5.          The setup tool presented in this work does not provide much novelty as it is very similar to the tools offered by the manufacturer, namely **TopSolids** and **bioTop**. The latter is largely for solution-state experiments, but the functionalities, such as creating experimental templates, is provided as well as an automated calibration function, and, I believe, solid-state setups are being currently included. Furthermore, **TopSpin** already provides a large and growing number of solid-state pulse sequences for biomacromolecules, but also for materials, which is not included in the library compiled by the authors. In my opinion, it would be more helpful to the NMR community to push the manufacturer to improve the tools that they already have to the desires of the users and provide them with state-of-the-art pulse sequences.

In principle we agree that it would be best if Bruker did this work, or if they integrated this library, and ensured its continuity over time. We have indeed discussed with Bruker, and while they are generally interested, it is a very slow and time-consuming process to get Bruker practically involved in such a development.

As an aside, currently Topsolids lacks largely the 1H-detected bio-ssNMR experiments, currently.

While it is not clear how Bruker's tools will evolve, we think that it makes sense to just move forward and make this tool freely available to the academic community.

In conclusion, it is a nice attempt and the authors introduce some new assignment experiments, however, the setup is only a minor advancement compared to already existing tools. In my view, joining forces with Bruker would be much more fruitful in terms of an universal solution for the NMR community.

We thank the reviewer for the overall positive assessment.

**Anonymous Referee #2**

The manuscript by Vallet et al introduces a library of pulse sequences for Bruker spectrometers that enables the users to easily implement a wide range of experiments commonly employed for resonance assignments or probing dynamics in solid state NMR.

While ssNMRlib is not a novel experimental technique, it could be of great use for users with different levels of expertise in setting up solid-state NMR experiments. The features that I find useful are:

(1) Easy optimisation protocol for coherence transfer steps that are the building blocks for complex resonance assignment experiments. (2) The ease with which a new experiment can be added to the library. (3) The ease with which the experimental parameters can be retrieved, especially RF powers in kHz units.

I would like to ask the following questions to the authors:

(1) The authors implement a security feature that adds one more layer to the "Power Check" feature of Topspin, which is concerned with duration. A full proof security feature is probably difficult to set up, but could the authors please comment about switching to a "duty cycle" based security system?

There a multiple ways, how one could implement security checks for a given pulse sequence, and all have their pros and cons. In the present version of ssNMRlib, we have opted for checking a range of pulse sequence parameters by comparing them to values listed in a file (see my reply to reviewer 1). In the future, we may also exploit the possibility of implement security features based on duty cycle calculations.

(2) As the authors promise, indeed it would be great if on-the fly shape generation could be implemented. It would also be helpful if optimal control derived shapes are available in ssNMRlib. However, judging from the setup of ssNMRlib, this can be achieved with reasonable efforts also from the users.

WaveMaker is part of the NMRlib distribution, and it has been used for a number of solution state NMR pulse sequences. Actually, when a new data set is loaded, WaveMaker is automatically executed, which makes it easy to use (or not) wavemaker commands in new pulse sequences.

For ssNMRlib, we have considered using WaveMaker to create shapes. WaveMaker would have one advantage over the current implementation: it would allow to calculate shapes on the fly, including arbitrary shapes, e.g. tangential CP ramps. WaveMaker would allow doing e.g. a popt optimization of the shape file directly within one popt run, rather than by comparing different shapes in different experiments.

We ended up deciding against implementing WaveMaker at the current stage. One reason is that we would have to go back and change some 140 pulse sequences; we decided for now not to do it and rather have a functional library in place. The second reason is that we wanted to have a coherent naming convention, where each type of CP (e.g. H-N, H-C, etc) would have the same name across all experiments. This means that we have almost all of the available constant names (cnst) assigned to CPs or other parameters. If we wanted to use WaveMaker, we would need to have additional parameters, such as the adiabaticity. The number of constants in Topspin is limited to 64 (cnst0 to cnst63). We cannot fit all constants that we would need within these 64. This is not a good reason, and we hope that Topspin will alleviate this limitation soon.

 (3) Like any other software, it would be great to have a "Troubleshooting" section that would help the beginners to address problems. In this regard, also an email discussion group among users would be beneficial.

This is a good idea. We have thought of a forum already, and will set it up soon.

(4) It is well accepted among NMR users that Topspin is not really well suited for performing quantitative data analysis. By integrating ssNMRlib with programs like nmrglue and matplotlib, the users can extend the workflow from setting up experiments to even reliable data analysis. Do the authors already plan to add features like this in ssNMRlib?

A number of macros in Topspin already use python scripts for data analysis (including nmrglue). For example, there are scripts in NMRlib which fit diffusion or overall-tumbling (TRACT) data. Likewise, scripts are available to export spectra as matplotlib figures.

These scripts are executed e.g. by clicking the cogwheel symbol right next to the DOSY or TRACT setup buttons in the solution-state library. Those examples indicate how to extract and export data to python/matplotlib.

We have added the following sentence: *"Further scripts, using python/nmrglue/numpy, allow extrating intensities, plotting spectra in matplotlib format or fit data, as discussed e.g. for solution-state NMR experiments (DOSY, TRACT etc), as described \citep{Favier2019}."*

To conclude, I find the manuscript interesting and publishable in Magnetic Resonance.

We thank the reviewer for the positive evaluation.

We have also done a small unsolicited change: we have stated that ssNMRlib has been tested on Topspin 3.5, 3.6 and 4, rather than "from 3.2 to 3.6 and 4". NMRlib has initially been developed on Topspin 3.2 pl7, but we do not know now if everything works on 3.2. It may, but we do not want to state it in the 
[revised manuscript text omitted]